# Constitutive Modeling of Physical Properties of Coastal Sand during Tunneling Construction Disturbance

**Jian-Feng Zhu [1], Hong-Yi Zhao [2],*, Ri-Qing Xu [3],*, Zhan-You Luo [1] and Dong-Sheng Jeng [4]**

1   School of Civil Engineering and Architecture, Zhejiang University of Science and Technology, Hangzhou 310023, China; zhujianfeng0811@163.com (J.-F.Z.); luozhanyou@nbu.edu.cn (Z.-Y.L.)
2   State Key Laboratory of Hydrology-Water Resources and Hydraulic Engineering, Hohai University, Nanjing 210098, China
3   Coastal and Geotechnical Research Center of Zhejiang University, Hangzhou 310058, China
4   Griffith School of Engineering, Griffith University Gold Coast Campus, Queensland, QLD 4222, Australia; dsjeng@icloud.com
*   Correspondence: hyzhao@hhu.edu.cn (H.-Y.Z.); xurq@zju.edu.cn (R.-Q.X.)

**Abstract:** This paper presents a simple but workable constitutive model for the stress–strain relationship of sandy soil during the process of tunneling construction disturbance in coastal cities. The model was developed by linking the parameter $K$ and internal angle $\varphi$ of the Duncan–Chang model with the disturbed degree of sand, in which the effects of the initial void ratio on the strength deformation property of sands are considered using a unified disturbance function based on disturbed state concept theory. Three cases were analyzed to investigate the validity of the proposed constitutive model considering disturbance. After validation, the proposed constitutive model was further incorporated into a 3D finite element framework to predict the soil deformation caused by shield construction. It was found that the simulated results agreed well with the analytical solution, indicating that the developed numerical model with proposed constitutive relationship is capable of characterizing the mechanical properties of sand under tunneling construction disturbance.

**Keywords:** sand; void ratio; disturbed state concept; disturbance function; constitutive model

## 1. Introduction

There is an ongoing demand for tunnel construction in coastal cities that need to shore up their crumbling infrastructure, seeking more efficient and less polluting modes of transportation. However, significant release of stress involved in the tunnel construction may cause catastrophic consequences for neighboring structures and underground works due to the excessive settlement and instability of the load-bearing soil layers. As reported by Chen et al. [1], the common lining uplift of Ningbo Metro Line 1 in eastern China during the tunneling construction stage reached more than 30 mm, which resulted in local cracks in tunnel linings and surrounding buildings. Consequently, estimation of potential ground movement during tunnel constructions is of great importance for civil engineers involved in the safe design of tunnels and their construction [2–6]. As a validated approach, finite element methods (FEMs) have been widely adopted to estimate the deformation characteristics of ground associated with complicated tunneling excavation [7–10]. To make predictions accurate, the essential features of soil behavior have to be reproduced by using suitable constitutive models with an FEM [11]. Addenbrooke et al. [12] developed a 2D FEM to investigate tunnel-induced ground movements in which the nonlinear behavior of soils is reproduced by adopting the Duncan–Chang model. Later, Zhang et al. [13] extended this framework to a 3D analysis of nailed soil structures under working loads. Mroueh and Shahrour [14] developed a full 3D finite element model to study the interaction between tunneling in soft soils and adjacent structures based on an elastic perfectly plastic constitutive relation with a Mohr–Coulomb criterion. Karakus and Fowell [15] utilized

the modified Cam-Clay model to investigate the effects of different excavation patterns on tunnel construction-induced settlement. Hejazi et al. [11] analyzed the impact of the Mohr–Coulomb model, the hardening soil (HS) model and the hardening soil model with small-strain stiffness (HS-Small) on the numerical analysis of underground constructions. In the aforementioned investigations, the stress–strain relationship for the soil continuum was idealized using linear elastic models; however, soil behavior during tunnel construction can never be purely elastic but always contains an elastoplastic element associated with the residual soil deformations due to excavation. However, the nonlinear elastic model is capable of capturing the nonlinearity, stress dependency and inelasticity of the soil behavior. Moreover, it has good convergence performance thanks to its elastic property [16,17].

Apart from the nonlinearity of soil deformation, the geotechnical engineer must also take into account factors caused by the construction disturbance. As shown in Figure 1, the disturbance of shield construction, which is one of the popular construction methods in coastal cities of China, affects the mechanical behavior of coastal sand by changing its physical properties, including the void ratio, water content and internal friction angle ($\varphi$) associated with the weakening of the initial tangent modulus ($E_i$), which may lead to uneven settlement and the cracking of nearby buildings. Such a complex disturbance process can be well reproduced using disturbed state concept (DSC) theory [18,19], in which the physical and mechanical behavior of structured geo-materials at any stage during deformation under mechanical and/or environmental loadings can be expressed in terms of the behavior of material parts in the two reference states: relatively intact (RI) and fully adjusted (FA) states. The deviation of the observed state from the RI (or FA) states is called disturbance and the observed behavior can be well-replicated through a disturbance function which couples the RI and FA states. Based on DSC theory, Liu et al. [18] proposed a unified model to predict the compression behavior of structured geo-materials including clay, sand, calcareous soil and gravel, which was extended to characterize the deformation performance of the metal-rich clays by Fan et al. [20]. Desai and EI-Hoseiny [19] and Zhu et al. [21] investigated the field response of reinforced soil walls and the earth pressure of rigid retaining walls. Pradhan and Desai [22] characterized the cyclic response of sands and interfaces between piles and sands by locating the critical disturbance during deformation. Zhu et al. [23] developed an analytical solution to predict shield construction-induced ground movements in green field by considering the disturbance effect of initial relative density on the shear modulus of sandy soil. Detailed information about DSC theory in applications for more materials and regions can be found in the work by Desai [24,25].

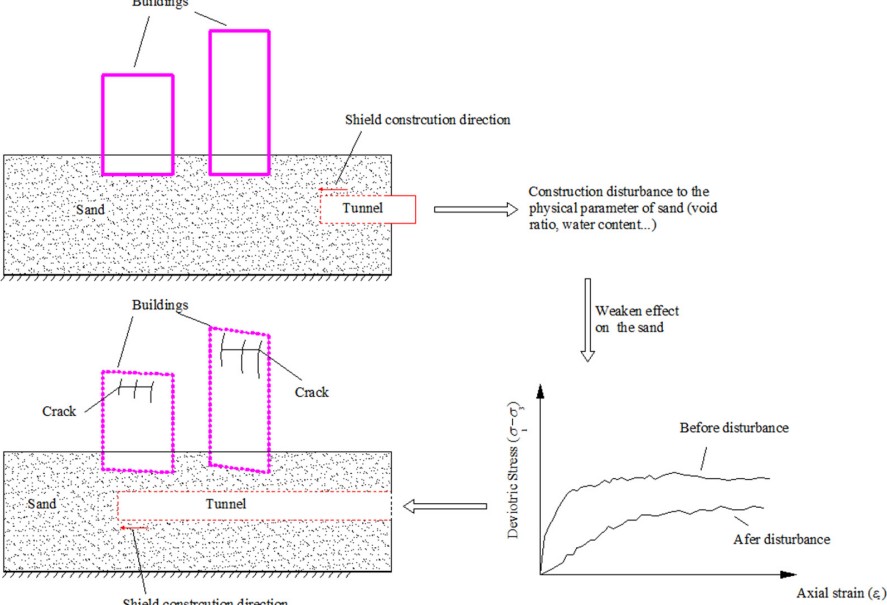

**Figure 1.** Shield construction disturbance to the mechanical properties of coastal sand.

In the present research, a series of triaxial compression tests were conducted for dry and saturated sands with different initial void ratios. The tested results were used to modify the disturbance function in terms of $K$ and $\varphi$, utilizing DSC theory. Based on the proposed disturbance function, a modified Duncan–Chang model [26] taking into account construction disturbance was established. The developed model was applied to reproduce the physical properties of another kind of sand in a disturbed state to identify the model's validity and effectiveness. The validated framework was further incorporated into a 3D finite element model to predict the soil deformation caused by shield construction.

## 2. Laboratory Test

The samples used in the tests were composed of ISO standard sand provided by the Xiamen Company of China. The particle size distribution (PSD) for the tested samples was within the range between 0.25 mm and 1 mm, as shown in Figure 2. The sample physical parameters are listed in Table 1 [23], where $G_s$ is the specific gravity; $e_{max}$ and $e_{min}$ are the maximum and minimum void ratio, respectively; $w$ is the water content; and $C_u$ and $C_c$ are the coefficients of uniformity and curvature, respectively.

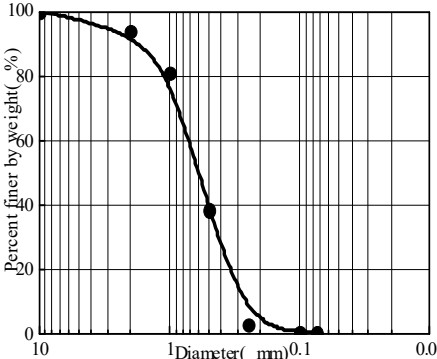

**Figure 2.** Particle size distribution curve of the ISO sample.

**Table 1.** Physical index of sands.

| Sand | $w/\%$ | $G_s$ | $e_{max}$ | $e_{mim}$ | $C_u$ | $C_c$ |
|------|--------|-------|-----------|-----------|-------|-------|
| ISO | 0.046 | 2.681 | 0.723 | 0.382 | 2.267 | 1.408 |

All the tests were performed in the geotechnical laboratory of Zhejiang University in China. The triaxial device used in this experiment was the SJ-1A model, designed and manufactured by Guo Dian Nanjing Automation Company Limited, China. In this apparatus, a servohydraulic system is applied to control the cyclic vertical stress and frequency of loading, whereas an oil pressure type piston is applied to control the confining pressure, which varies from 0 to 2 MPa.

The specimens were prepared with four different initial void ratios ($e_0$). The specific values for these specimens are listed in Table 2, in which $m$ and $m_a$ are parameters defining the mass of the specimen and the mass of each layer, respectively. The specimens in the tests were prepared following the techniques of the SL237-1999 standard [27] and tested at three different confining levels (100, 200 and 300 kPa) and a fixed value of $e_0$. A total of 24 standard undrained monotonic triaxial tests were conducted by Zhu et al. [23] at a strain rate of 0.808 mm/min for dry sand, with the failure criterion being controlled by the peak strength. The test results are shown in Figure 3, in which $\sigma_1$ and $\sigma_3$ are the principal stresses corresponding to the axial and circumferential directions in the test, respectively, and $\varepsilon_{1z}$ represents the axial strain. As can be seen, as the initial void ratio $e_0$ decreased, the stress–strain curve for any confining pressure became steeper and the peak strength increased. Moreover, the strain that corresponded to the peak strength for all tests was approximately within the range between 2% and 3%.

**Table 2.** Experimental cases for the ISO sand.

| $e_0$ | $m$/g | $\overline{m}$/g |
|---|---|---|
| 0.59 | 161.790 | 32.358 |
| 0.56 | 166.785 | 33.357 |
| 0.52 | 168.957 | 33.791 |
| 0.49 | 174.550 | 34.910 |

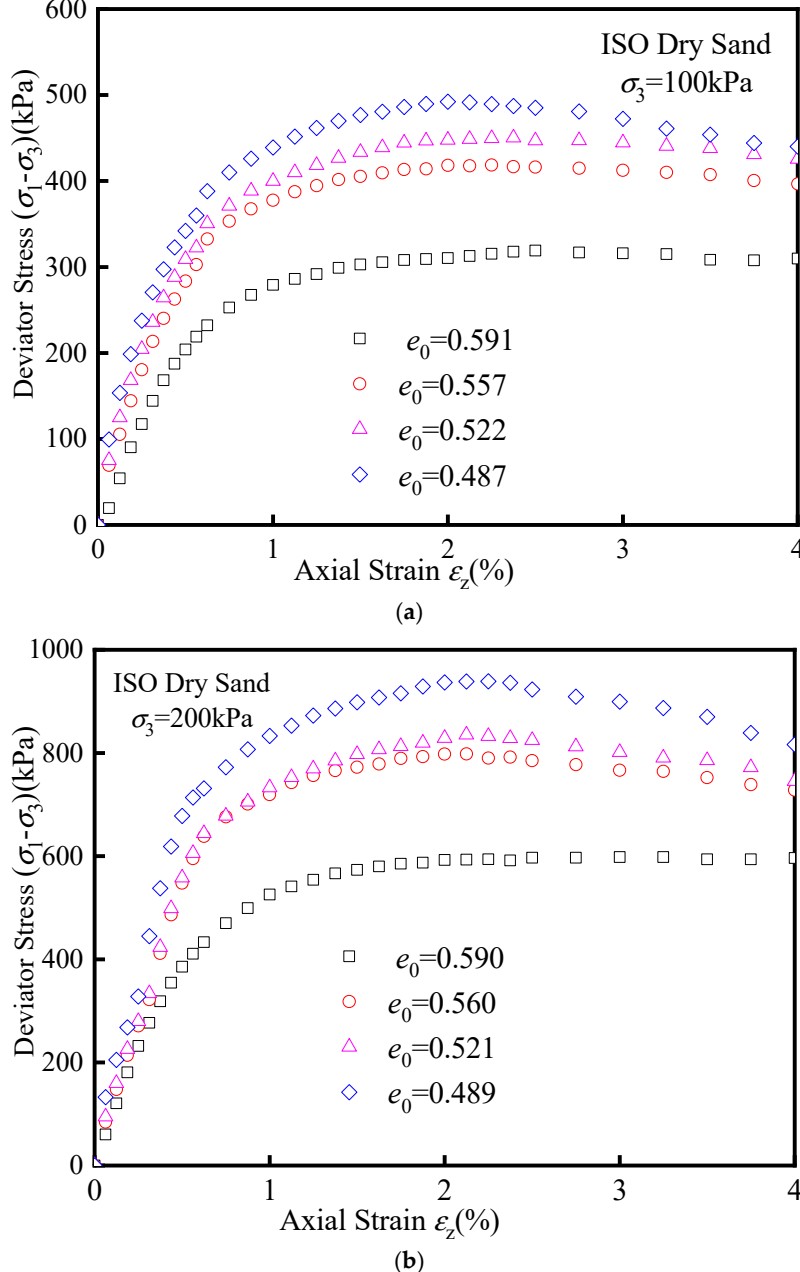

**Figure 3.** *Cont.*

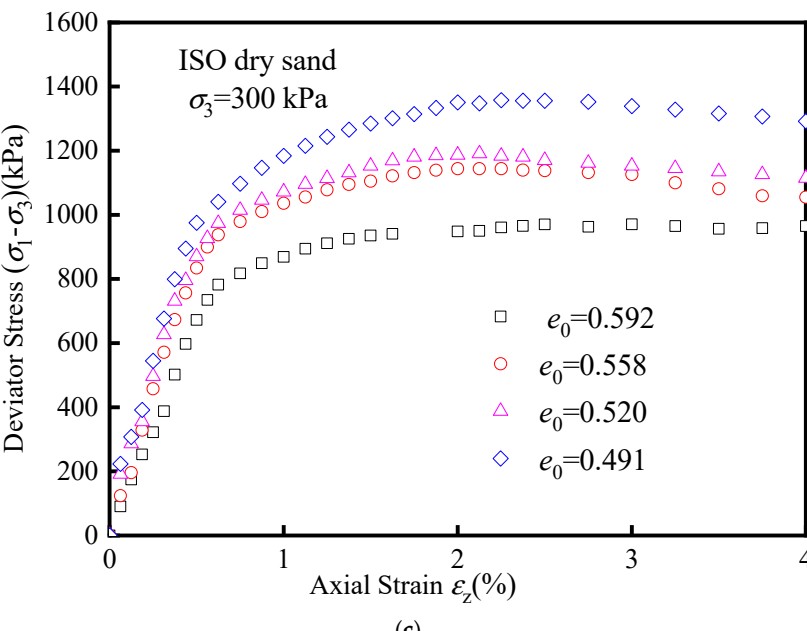

(**c**)

**Figure 3.** Deviator stress versus axial strain for the ISO dry sand in the undrained triaxial tests. (**a**) $\sigma_3$ = 100 kPa; (**b**) $\sigma_3$ = 200 kPa; (**c**) $\sigma_3$ = 300 kPa.

## 3. A Simplified Constitutive Model Considering Disturbance

### 3.1. Initial Tangent Modulus and Internal Friction Angle for Different Void Ratios

Kondner et al. [28,29] suggested that the nonlinear behavior of soils such as clay and sand can be effectively estimated with a hyperbola function, expressed as

$$\sigma_1 - \sigma_3 = \frac{\varepsilon_1}{a + b\varepsilon_1} \tag{1}$$

where *a* and *b* are model parameters for determining the initial tangent modulus and critical stress state when the stress–strain curve approaches infinite strain $(\sigma_1 - \sigma_3)_{\text{ult}}$.

Duncan and Chang [26] suggested that *a* and *b* could be determined as

$$a = \frac{1}{E_i} = \frac{\left(\frac{\varepsilon_1}{\sigma_1 - \sigma_3}\right)_{95\%} + \left(\frac{\varepsilon_1}{\sigma_1 - \sigma_3}\right)_{70\%}}{2} - \frac{\frac{\varepsilon_1}{(\sigma_1 - \sigma_3)_{\text{ult}}}[(\varepsilon_1)_{95\%} + (\varepsilon_1)_{70\%}]}{2} \tag{2}$$

$$b = \frac{1}{(\sigma_1 - \sigma_3)_{\text{ult}}} = \frac{\left(\frac{\varepsilon_1}{\sigma_1 - \sigma_3}\right)_{95\%} - \left(\frac{\varepsilon_1}{\sigma_1 - \sigma_3}\right)_{70\%}}{(\varepsilon_1)_{95\%} - (\varepsilon_1)_{70\%}} \tag{3}$$

where the subscripts 95% and 70% represent the ratio between the values of stress difference $(\sigma_1 - \sigma_3)$ and their peak values of strength $(\sigma_1 - \sigma_3)_f$, respectively.

The relationship between $(\sigma_1 - \sigma_3)_f$ and $(\sigma_1 - \sigma_3)_{\text{ult}}$ is established by the failure ratio $R_f$ as

$$R_f = \frac{(\sigma_1 - \sigma_3)_f}{(\sigma_1 - \sigma_3)_{\text{ult}}} \tag{4}$$

With the test results in Figure 3, the parameters $(\sigma_1 - \sigma_3)_f$, *a*, *b*, $E_i$, $(\sigma_1 - \sigma_3)_{\text{ult}}$ and instant failure ratio $R_{fi}$ can be determined accordingly by Equations (2)–(4). The results are shown in Table 3.

**Table 3.** Parameters of the ISO dry sands for each case.

| $\sigma_3$ /MPa | $e_0$ | $(\sigma_1 - \sigma_3)_f$ /MPa | $a$ /MPa$^{-1}$ ($\times 10^{-3}$) | $E_i$ /MPa | $(\sigma_1 - \sigma_3)_{ult}$ /MPa | $b$/ MPa$^{-1}$ | $R_{fi}$ |
|---|---|---|---|---|---|---|---|
| 0.1 | 0.591 | 0.319 | 11.223 | 89.100 | 0.392 | 2.553 | 0.814 |
| | 0.557 | 0.419 | 8.029 | 124.553 | 0.526 | 1.900 | 0.797 |
| | 0.522 | 0.450 | 7.119 | 140.471 | 0.547 | 1.827 | 0.823 |
| | 0.487 | 0.492 | 6.264 | 159.651 | 0.598 | 1.671 | 0.823 |
| 0.2 | 0.590 | 0.598 | 6.290 | 158.995 | 0.762 | 1.313 | 0.785 |
| | 0.560 | 0.792 | 4.160 | 240.401 | 1.014 | 0.986 | 0.7815 |
| | 0.521 | 0.835 | 3.801 | 263.118 | 1.000 | 1.000 | 0.835 |
| | 0.489 | 0.936 | 2.844 | 351.644 | 1.086 | 0.921 | 0.862 |
| 0.3 | 0.592 | 0.970 | 3.152 | 317.219 | 1.176 | 0.851 | 0.825 |
| | 0.558 | 1.144 | 2.428 | 411.843 | 1.359 | 0.736 | 0.842 |
| | 0.520 | 1.191 | 2.251 | 444.191 | 1.389 | 0.720 | 0.857 |
| | 0.491 | 1.358 | 2.017 | 495.719 | 1.581 | 0.633 | 0.859 |

Based on the laboratory tests, Janbu [30] suggested that the relationship between the initial tangent modulus and the confining pressure could be expressed as

$$E_i = Kp_a \left( \frac{\sigma_3}{p_a} \right)^n \tag{5}$$

where $p_a$ is the atmospheric pressure, $K$ is a model constant and $n$ is a dimensionless parameter related to the rate of variation of $E_i$ and $\sigma_3$.

Another form of Equation (5) can be stated as

$$\lg(E_i/p_a) = \lg K + n(\sigma_3/p_a) \tag{6}$$

With regard to the Mohr–Coulomb failure criterion, the peak failure strength can be derived as [26–31]

$$(\sigma_1 - \sigma_3)_f = \frac{2c \cos \phi + 2\sigma_3 \sin \phi}{1 - \sin \phi} \tag{7}$$

where $c$ and $\varphi$ are the cohesion and friction angle of the soil, respectively. Generally, $c = 0$ for sand, so the value $\varphi$ can be derived as

$$\phi = \arcsin \left[ \frac{(\sigma_1 - \sigma_3)_f}{2\sigma_3 + (\sigma_1 - \sigma_3)_f} \right] \tag{8}$$

Then, the values of the parameters $K$, $n$ and the instant internal friction angle $\varphi_i$ can be determined based on the experimental results in combination with Equations (6) and (8). The results are shown in Table 4, where $\varphi$ and $R_f$ are the mean values of $\varphi_i$ and $R_{fi}$ associated with each confining pressure.

Basically, the parameters $n$ and $R_f$ are not sensitive to the variation of $e_0$ defining the tunnel disturbance during the test. Therefore, only the relationships between $K$, $\varphi$ and $e_0$ are provided, as shown in Figures 4 and 5, respectively. The test results show that both $\ln K$ and $\sin \varphi$ are linearly proportional to the change of $e_0$. Hence the relationship between $K$, $\varphi$ and $e_0$ can be specified as follows:

$$\ln K = d + fe \tag{9}$$

$$\sin \phi = g + he \tag{10}$$

where $d$, $f$, $g$ and $h$ are dimensionless parameters which can be effectively determined using linear regression of the test results with correlation coefficients greater than 0.92.

**Table 4.** Parameters of the ISO dry sands for each case.

| $e_0$ | $\sigma_3$ /MPa | $\varphi_i$ /(°) | $\varphi$ /(°) | $R_f$ | $K$ | $n$ |
|---|---|---|---|---|---|---|
| | 0.1 | 37.927 | | | | |
| 0.59 | 0.2 | 36.818 | 37.636 | 0.808 | 952.094 | 0.919 |
| | 0.3 | 38.163 | | | | |
| | 0.1 | 39.868 | | | | |
| 0.56 | 0.2 | 40.022 | 39.785 | 0.807 | 1342.054 | 0.907 |
| | 0.3 | 39.465 | | | | |
| | 0.1 | 42.585 | | | | |
| 0.52 | 0.2 | 41.637 | 41.737 | 0.838 | 1495.099 | 0.886 |
| | 0.3 | 40.989 | | | | |
| | 0.1 | 43.829 | | | | |
| 0.49 | 0.2 | 42.546 | 42.685 | 0.848 | 1776.847 | 0.891 |
| | 0.3 | 41.681 | | | | |

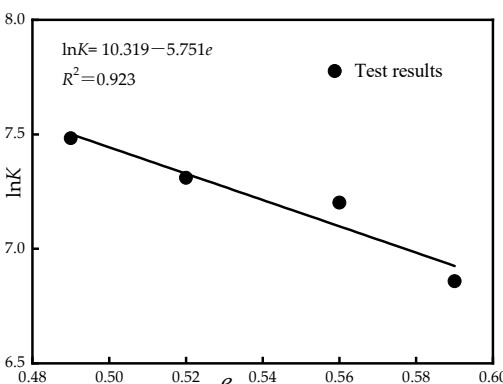

**Figure 4.** Relationship between $K$ and $e$.

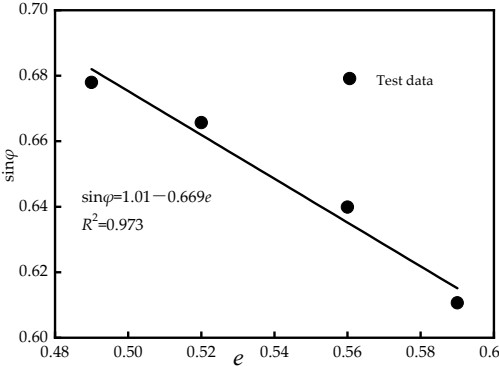

**Figure 5.** Relationship between $\varphi$ and $e$.

$E_i$ can be determined by substituting Equation (9) into Equation (5) as

$$E_i = \exp(d + fe)\, p_a (\sigma_3 / p_a)^n \tag{11}$$

The parameter $(\sigma_1 - \sigma_3)_f$ defining the critical stress state of soil should also be modified by substituting Equation (10) into Equation (7), with a simplified expression, as

$$(\sigma_1 - \sigma_3)_f = 2\sigma_3 \frac{g + he}{1 - (g + he)} \tag{12}$$

### 3.2. Unified Disturbance Function

In this study, a generalized disturbance function $D$ varying from $-1$ to $1$ was used to determine the degree of soil disturbance during the tunnel construction, as shown in Figure 6. Basically, there is no disturbance in soil when it is at the initial void ratio state. As the soil becomes looser, the void ratio $e$ increases, whereas the corresponding degree of disturbance ($D$) decreases. When $e$ approaches $e_{max}$, $D$ approaches $-1$ and the sand arrives at the loosest state. In turn, as the sand becomes denser, the amount of $e$ decreases, whereas $D$ increases. When $e$ approaches $e_{min}$, $D$ reaches 1 and the backfill arrives at the densest state. The relationship between the disturbed degree and void ratio can be expressed as below.

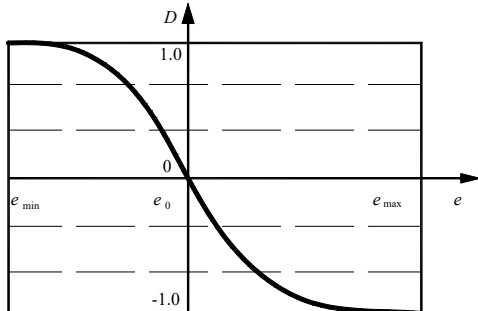

**Figure 6.** Relationship between the degree of disturbance and the void ratio.

(1)   For "positive disturbance" ($e \leq e_0$)

$$D = \frac{2}{\pi}\arctan\left(\frac{e_0 - e}{e - e_{min}}\right) \tag{13a}$$

(2)   For "negative disturbance" ($e > e_0$)

$$D = \frac{2}{\pi}\arctan\left(\frac{e_0 - e}{e_{max} - e}\right) \tag{13b}$$

### 3.3. Simplified Constitutive Model Considering Disturbance

(1) For "positive disturbance" ($e \leq e_0$), Equation (13a) can be rewritten as

$$e_0 - e = (e - e_{min})\tan(\pi D/2) \tag{14}$$

Substituting Equation (14) into Equation (9), the parameter $K_D$ during the disturbance can be expressed as

$$K_D = \exp\{d + f[e_0 - (e - e_{min})\tan(\pi D/2)]\} \tag{15}$$

Substituting Equation (14) into Equation (10), the parameter $\varphi_D$ during disturbance can be expressed as

$$\sin\phi_D = g + h[e_0 - (e - e_{min})\tan(\pi D/2)] \tag{16}$$

Substituting Equations (4), (15) and (16) into Equation (1), the relationship between the deviator stress and axial strain considering "positive disturbance" can be expressed as

$$\sigma_1 - \sigma_3 = \frac{\varepsilon_1}{\frac{1}{\exp\{d+f[e_0-(e-e_{\min})\tan(\pi D/2)]\}p_a(\sigma_3/p_a)^{n_0}} + \frac{1-\{g+h[e_0-(e-e_{\min})\tan(\pi D/2)]\}R_{f0}}{2\sigma_3\{g+h[e_0-(e-e_{\min})\tan(\pi D/2)]\}}\varepsilon_1} \tag{17}$$

(2) For "negative disturbance" ($e > e_0$), Equation (13b) can be rewritten as

$$e_0 - e = (e_{\max} - e)\tan(\pi D/2) \tag{18}$$

Substituting Equation (18) into Equation (9), the parameter $K_D$ during the disturbance can be expressed as

$$K_D = \exp\{d + f[e_0 - (e_{\max} - e)\tan(\pi D/2)]\} \tag{19}$$

Substituting Equation (18) into Equation (10), the parameter $\varphi_D$ during the disturbance can be expressed as

$$\sin\phi_D = g + h[e_0 - (e_{\max} - e)\tan(\pi D/2)] \tag{20}$$

Substituting Equations (4), (19) and (20) into Equation (1), the relationship between deviator stress and axial strain considering "negative disturbance" can be expressed as

$$\sigma_1 - \sigma_3 = \frac{\varepsilon_1}{\frac{1}{\exp\{d+f[e_0-(e_{\max}-e)\tan(\pi D/2)]\}p_a(\sigma_3/p_a)^{n_0}} + \frac{1-\{g+h[e_0-(e_{\max}-e)\tan(\pi D/2)]\}R_{f0}}{2\sigma_3\{g+h[e_0-(e_{\max}-e)\tan(\pi D/2)]\}}\varepsilon_1} \tag{21}$$

A total of twelve parameters, namely $e_0$, $e_{\max}$, $e_{\min}$, $\overline{R}_{f0}$, $K_0$, $n_0$, $\varphi_0$, $\sigma_3$, $d$, $f$, $g$ and $h$, are covered in the proposed model. Among these, $e_0$, $e_{\max}$ and $e_{\min}$ can be easily determined by the fundamental physical test of sand; $\varphi_0$, $k_0$, $\overline{R}_{f0}$, $n_0$, are the same as in the original Duncan–Chang model; and $d$, $f$, $g$ and $h$ can be calibrated by the traditional undrained triaxial tests of sand.

## 4. Verification

We next took another kind of dry sand, Fujian standard sand (FJ sand), as the test material and conducted a series of triaxial compression tests to assess the validity of the proposed simplified constitutive model for disturbed states. The particle size distributions of the Fujian standard sand mainly ranged from 0.25 mm to 1 mm, as shown in Figure 7 [23]. Its physical parameters are listed in Table 5. The associated parameters of the proposed simplified constitutive model are listed in Table 6. Suppose that the initial void ratio $e_0$ of the sand is 0.76 and the sands at other void ratios (e.g., $e = 0.79, 0.73, 0.70$) are at different disturbed states, with their corresponding disturbed degrees as shown in Table 7. The predicted results of the proposed model are shown in Figure 8a–d. It can be seen that the stress–strain relationship of the sandy soil was significantly affected, either positively or negatively, by the disturbances. The predicted results always agreed well with the test curve at any disturbed state.

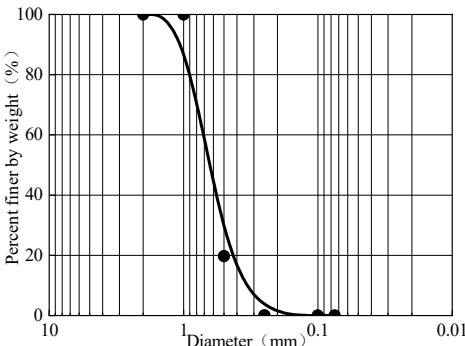

**Figure 7.** Particle size distribution curve of the Fujian standard sand (FJ sand) sample.

**Table 5.** Physical index of FJ sand.

| Sand | $w$ /% | $G_s$ | $e_{max}$ | $e_{mim}$ | $C_u$ | $C_c$ |
|---|---|---|---|---|---|---|
| FJ | 0.045 | 2.697 | 0.926 | 0.645 | 1.442 | 0.923 |

**Table 6.** Parameters of the simplified constitutive model.

| $e_0$ | $\overline{R}_{f0}$ | $K_0$ | $n_0$ | $\varphi_0$ /(°) | $\sigma_3$ / kPa | $d$ | $f$ | $g$ | $h$ |
|---|---|---|---|---|---|---|---|---|---|
| 0.76 | 0.826 | - | 0.881 | - | 100 | 200 | 300 | 12.311 | −6.956 |

**Table 7.** Soil disturbance degree of the FJ dry sand.

| $e$ | $D$ |
|---|---|
| 0.79 | −0.138 |
| 0.76 | 0.000 |
| 0.73 | 0.216 |
| 0.70 | 0.528 |

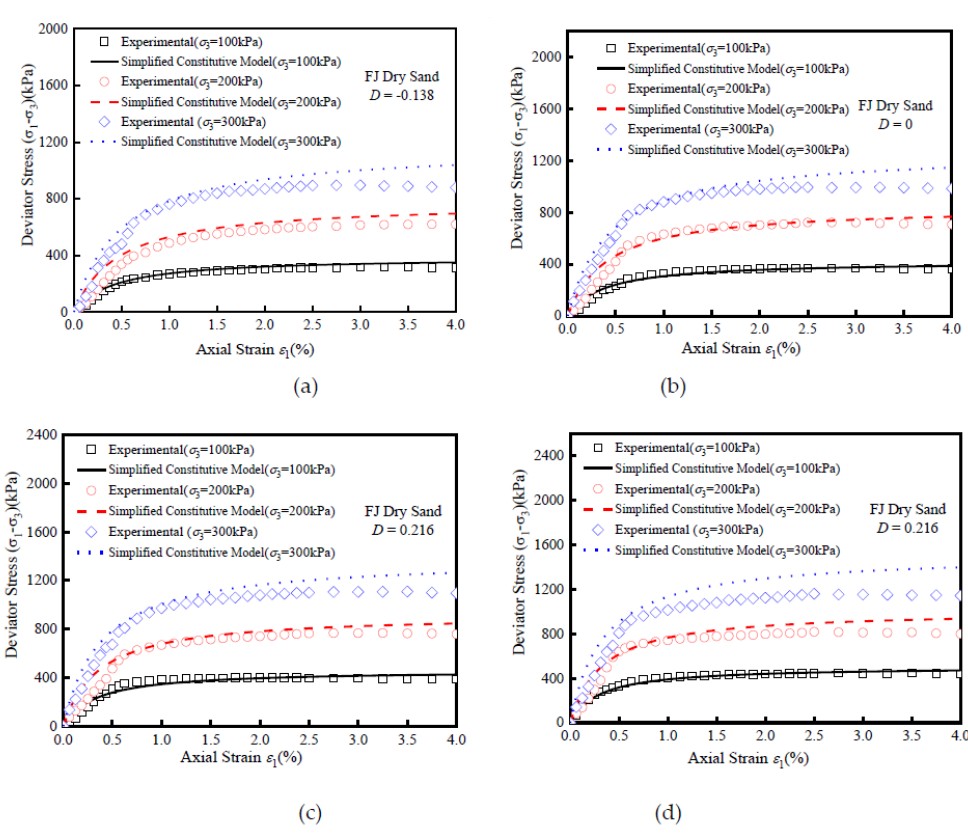

**Figure 8.** Theoretical and experimental stress–strain curves for the FJ dry sand. (**a**) $D = -0.318$; (**b**) $D = 0$; (**c**) $D = 0.216$; (**d**) $D = 0.528$.

## 5. Application

The developed constitutive framework was further incorporated into the commercially available software ABAQUS to reproduce the ground movement during tunnel constructions and compare the acquired simulation with the analytical solution.

In this study, a metro tunnel was considered as being 26.2 km long, 6.2 m in diameter and 0.35 m thick. The computing parameters of the tunnel are listed in Table 8. Assuming that the soil was of isotropic and homogeneous behavior, only half of the whole tunnel

(Figure 9) was modeled to optimize the computational cost. The FEM mesh consisted of 10,010 nodes and 9480 elements (six- and eight-node linear brick), conditioned with appropriate boundary conditions. The upper surface corresponding to the effective ground was free to move, for which the pressure induced by the self-gravity of the EPB-S machine was taken into account by applying a constant distributed load equal to 20 kPa [1,32] (additional thrust *p*). The interactive behavior of the shield–soil wall due to the effects of fluid injections from the shield head was conditioned with an additional friction force $\tau$ (45 kPa) [33,34]. The physical and mechanical parameters of the soil for the analytical solution [23] are reported in Table 9.

**Table 8.** Computing parameters of the tunnel.

| *R* /m | *H* /m | *L* /m |
|---|---|---|
| 3.195 | 11.848 | 9.00 |

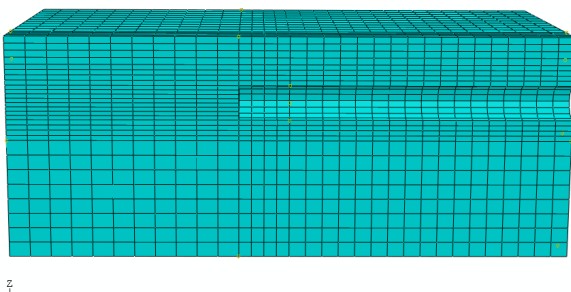

**Figure 9.** Mesh division diagram of the computed model.

**Table 9.** Computing parameters of soil for the analytical method.

| $\nu$ | $E_{S0}$ /kPa | $D_{r0}$ | $D_{rmin}$ | $D_{rmax}$ | *d* | *p* /kPa | $\tau$ /kPa |
|---|---|---|---|---|---|---|---|
| 0.30 | 17,000.00 | 0.50 | 0.00 | 1.00 | 1.933 | 20.00 | 45.00 |

The soil was considered to be isotopically nonlinearly elastic and homogeneous, with Poisson's ratio constant during the shield construction disturbance. The additional thrust *p* and additional friction force $\tau$ were also assumed to be acting straight on the soil in contrast with the analytical solution [23]. The physical properties assumed for the soils are reported in Table 10.

**Table 10.** Computing parameters of soil for constitutive model considering disturbance.

| $\gamma$ /kN/m$^3$ | $\nu$ | $e_0$ | $e_{min}$ | $e_{max}$ | $\overline{R}_{f0}$ | $n_0$ | *d* | *f* | *g* | *h* |
|---|---|---|---|---|---|---|---|---|---|---|
| 19 | 0.3 | 0.502 | 0.362 | 0.646 | 0.806 | 0.9 | 1.933 | 5.846 | 4.947 | −8.4 |

Figures 10 and 11 show the stress and vertical displacement fields before additional thrust and additional friction force act upon the soil. The initial stress field is well balanced because the stress of the soil is in line with the depth and the maximum vertical displacement is $10^{-7}$ mm.

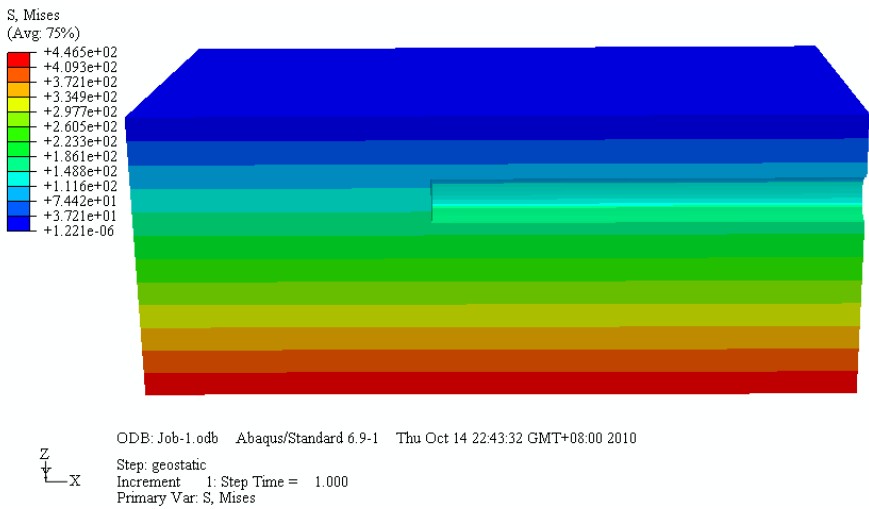

**Figure 10.** Von Mises stress clouds of the initial state (kPa).

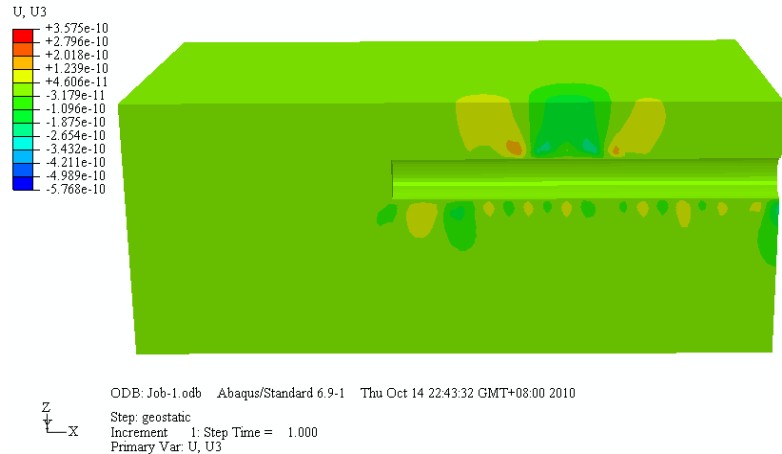

**Figure 11.** Vertical ground movement clouds of the initial state (m).

Figures 12 and 13 show the stress and vertical displacement fields after the additional thrust acts on the system. The vertical displacements before and behind the shield area show eminence and subsidence, respectively. However, there is no significant change in the stress field of the soil.

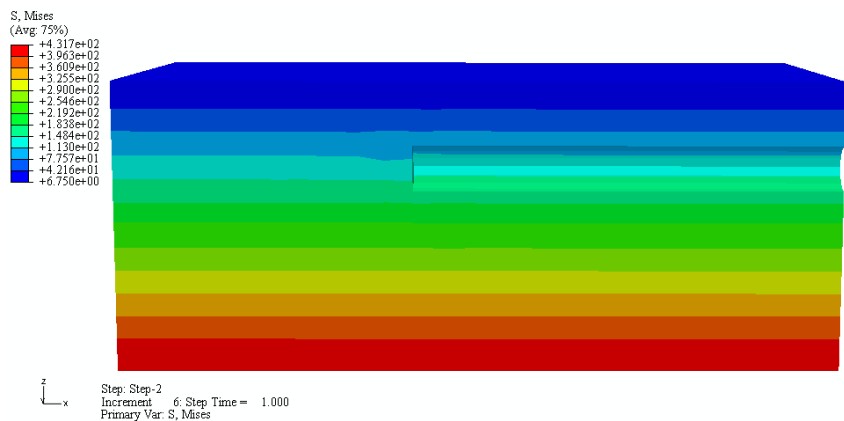

**Figure 12.** Von Mises stress clouds of the model under additional thrust action (kPa).

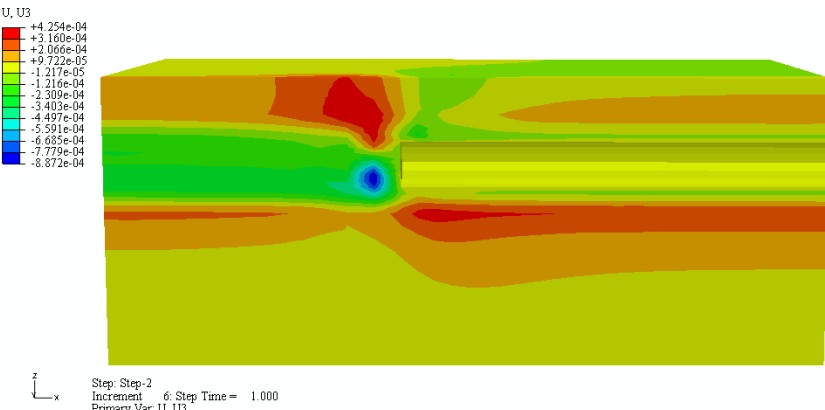

**Figure 13.** Vertical ground movement clouds of the model under additional thrust action (m).

Figures 14 and 15 show the stress and vertical displacement fields after the additional friction force acts on the system. The vertical displacements before and behind the shield area also show eminence and subsidence, respectively. Moreover, there is significant change in the stress field of the soil.

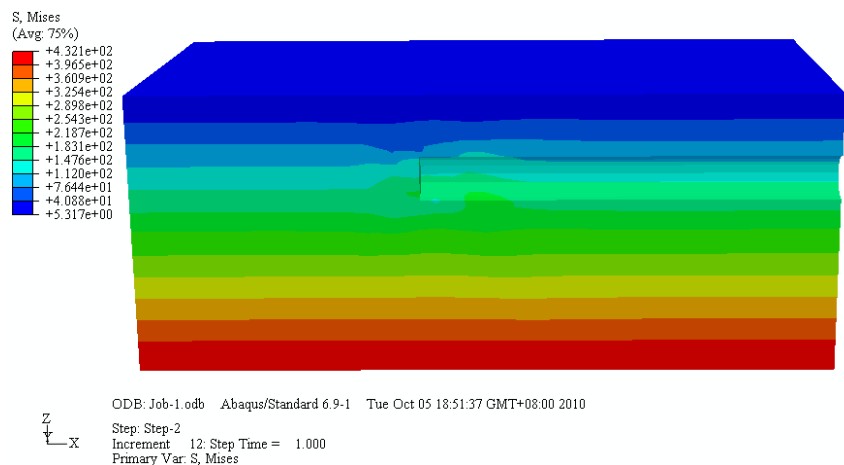

**Figure 14.** Von Mises stress clouds of the model under additional friction force (kPa).

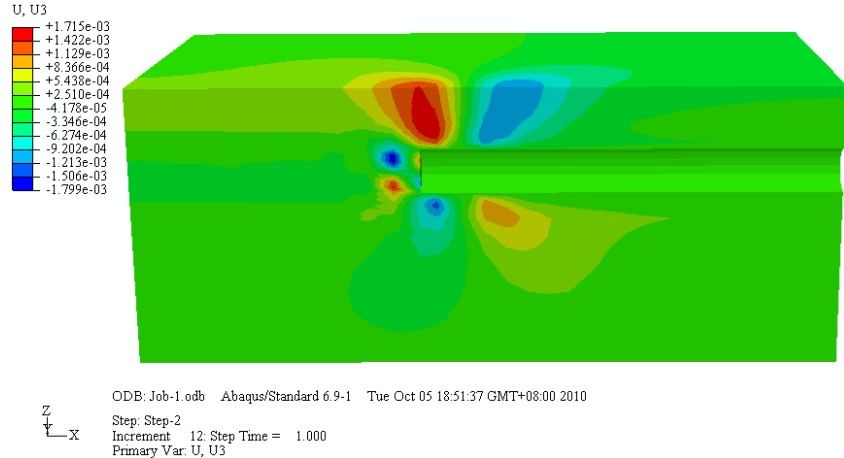

**Figure 15.** Vertical ground movement clouds of the model under additional friction force (m).

During shield construction, a uniformed change in the relative density of the soil is often considered. Different states of soil around the tunnel during shield construction can then be covered by assuming five different relative soil densities, in which $D_r = 0.5$

is the initial state, $D_r$ = 0.3 and 0.4 are the negative disturbed states, and $D_r$ = 0.6 and 0.7 are the positive disturbed states. The corresponding degree of disturbance computed by Equation (14) for each case is shown in Table 11. The additional thrust and friction force at the different disturbed states can be calculated using the 3D finite element model, taking into account ground movements and considering disturbance. As a comparison, the results based on the analytical solution proposed by Zhu et al. [23] are incorporated in Figures 16 and 17. As can be seen, the predicted vertical ground movement of the 3D FEM always agreed well with the analytical solution in any disturbed state. The observations in Figures 10–17 indicate that the proposed constitutive model can characterize the mechanical properties of sand under construction disturbance.

**Table 11.** Soil disturbance degree in different cases.

| $e$ | $D_r$ | $D$ |
|---|---|---|
| 0.561 | 0.3 | −0.386 |
| 0.532 | 0.4 | −0.164 |
| 0.502 | 0.5 | 0.000 |
| 0.476 | 0.6 | 0.143 |
| 0.447 | 0.7 | 0.366 |

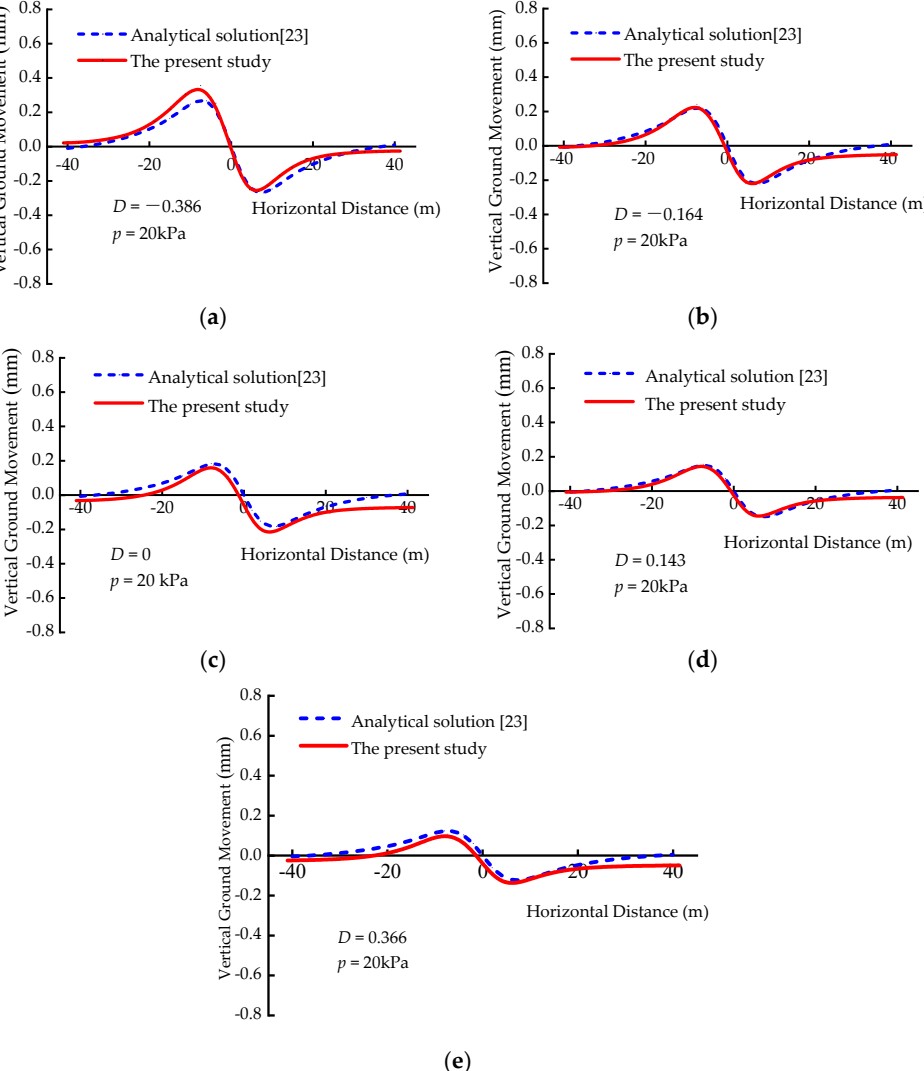

**Figure 16.** Vertical ground movements due to additional thrust at different disturbed states. (**a**) $D$ = −0.386; (**b**) $D$ = −0.164; (**c**) $D$ = 0; (**d**) $D$ = 0.143; (**e**) $D$ = 0.366.

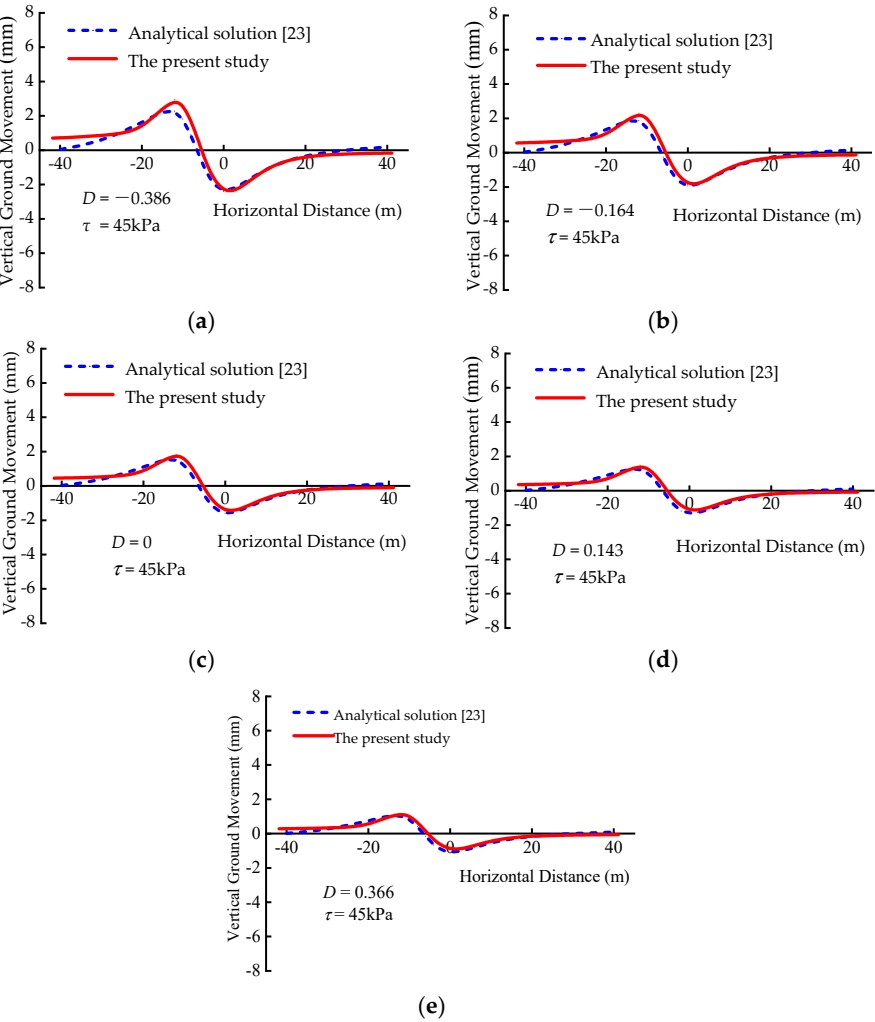

**Figure 17.** Vertical ground movements due to additional friction force at different disturbed states. (a) $D = -0.386$; (b) $D = -0.164$; (c) $D = 0$; (d) $D = 0.143$; (e) $D = 0.366$.

## 6. Conclusions

The motivation of the present study was to develop a practical constitutive model with the capacity to predict nonlinear soil behavior during the tunneling construction disturbance process. First, a series of undrained triaxial tests were conducted for samples of ISO standard sand with different initial void ratios. It was found that both the slope of the stress–strain curve and the peak strength increase when the value of $e_0$ decreases. Based on the test results, a unified disturbance function was proposed based on DSC theory, in which the void ratio was selected as the disturbance parameter. Then, a novel approach relating the parameter $K$ and internal angle $\varphi$ to the disturbed degree was derived to modify the constitutive model considering construction disturbance. The proposed constitutive model was used to predict the physical properties of Fujian standard sand in a disturbed state. The results show that the predicted results for the stress–strain relationship of the soil always agreed well with the experimental data for any disturbed state. Finally, the proposed constitutive model was incorporated into the finite element modeling and validated against the analytical solution [23]. The results show that the predicted results in terms of vertical ground movements were in good agreement with the analytical solution [23] for any disturbed state, indicating that the developed model is capable of reproducing the mechanical behavior of sandy soil across the whole process of shield construction and can be extensively implemented for predicting construction disturbance effects in practical tunneling engineering.

It should be noted here that the proposed disturbance function is fundamental and does not cover physical parameters such as water content, mass density, etc. Moreover, the inherent relationship between the mechanical parameters, such as stiffness, cohesion and internal friction angle, of clay or sand-clay admixture and the aforementioned physical parameters under the disturbed state of tunneling construction should be further addressed. More advanced solutions including these parameters will be further studied in future research.

**Author Contributions:** Methodology and Writing—original draft, J.-F.Z.; Software, H.-Y.Z.; Supervision, R.-Q.X.; Writing—Review and Editing, Z.-Y.L. and D.-S.J. All authors have read and agreed to the published version of the manuscript.

**Funding:** This research was funded by the National Natural Science Foundation of China (51879133, 51909077, 41572298).

**Institutional Review Board Statement:** Not applicable.

**Informed Consent Statement:** Not applicable.

**Data Availability Statement:** Not applicable.

**Conflicts of Interest:** The authors declare no conflicts of interest.

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
