# Peer review of "Constitutive Modeling of Physical Properties of Coastal Sand during Tunneling Construction Disturbance"

_jmse, doi:10.3390/jmse9020167_

Round 1

Reviewer 1 Report

I found that manuscript elegantly prepared and in excellent English. The specific contributions of the authors are specified. With the caveat that I do not claim expertise in geotechnical theory, I have only minor suggestions, questions, and a few stylistic points.

  1. I think it would be helpful to refer to specific areas in China where tunnel projects of the indicated kind are underway.
  2. The current approach is limited to sands. It is known that clay admixtures can affect the angle of internal friction in constitutive models. It would extend understanding of the applicability of the modifications of the Duncan-Chang model to provide discussion of how the stress-strain relationship might be affected in the presence of clay.
  3. It might be helpful to clarify how the current model “simplifies” applications
  4. Minor points: after reference to Zhu [23] “green filed”?; clarify “shield construction”; in Fig. 3 increase font for the label of horizontal axis; given the importance of equation (21) increase font size?; the term nephogram was not familiar to me in this context (normally it refers to clouds) – perhaps clarify in parenthesis? Fig. 17 fonts for symbols are extremely small and hard for people with reduced vision to see.

Author Response

Reviewer1: I found that manuscript elegantly prepared and in excellent English. The specific contributions of the authors are specified. With the caveat that I do not claim expertise in geotechnical theory, I have only minor suggestions, questions, and a few stylistic points.

Comment #1-1: I think it would be helpful to refer to specific areas in China where tunnel projects of the indicated kind are underway.

Response #1-1: The authors agree with the reviewer’s advice. In the revised manuscript, one case has been added as follows[Page 1, Lines 33-36].

….As reported by Chen et al.[1], the common lining uplift of Ningbo Metro Line 1 in eastern China during the tunneling construction stage reached more than 30 mm, which resulted in local cracks of tunnel linings and surrounding buildings. Consequently,…

Comment #1-2: The current approach is limited to sands. It is known that clay admixtures can affect the angle of internal friction in constitutive models. It would extend understanding of the applicability of the modifications of the Duncan-Chang model to provide discussion of how the stress-strain relationship might be affected in the presence of clay.

Response #1-2: This is a fair comment. The authors believe that the presence of clay will not only have an influence on the internal friction angle of sand but also improve the cohesion of sand. In future studies, the authors will consider how to develop new constitutive model to reproduce the mechanical behaviour of sand-clay admixtures during the process of tunneling construction disturbance. The following sentences are added in the revised manuscript as follows [Page 16, Lines 26-29].

….Moreover, the inherent relationship between the mechanical parameters such as stiffness, cohesion, internal friction angle, etc. of clay or sand-clay admixture and the aforementioned physical parameters under the disturbed state of tunneling construction should be further addressed….

Comment #1-3: It might be helpful to clarify how the current model “simplifies” applications.

Response #1-3: As the reviewer suggested, in the revised manuscript the authors have added the description to demonstrate how straightforwardly the parameters can be calibrated in the developed model to facilitate its application for practical engineering design. [Page 10, Lines 3-6].

…. A total of twelve parameters, namely, e0, emax, emin, , K0, n0, φ0, σ3, d, f, g and h are covered in the proposed model. Among these, e0, emax and emin can be easily determined by the fundamental physical test of sand, φ0, k0,, n0, are is the same as the original Duncan–Chang model, d, fg and h can be calibrated by the traditional undrained triaxial tests of sand….

Comment #1-4: Minor points: after reference to Zhu [23] “green filed”?; clarify “shield construction”; in Fig. 3 increase font for the label of horizontal axis; given the importance of equation (21) increase font size?; the term nephogram was not familiar to me in this context (normally it refers to clouds) – perhaps clarify in parenthesis? Fig. 17 fonts for symbols are extremely small and hard for people with reduced vision to see.

Response #1-4: Thanks for the reminding. The authors have gone through the manuscript and the associated errors have been corrected as follows:

after reference to Zhu [23] “green filed”?

In Page 3 Line 7, ‘green filed’ has been corrected to be “green field”

clarify “shield construction”

In Page 2 Lines 17-18, the following sentence have been added to clarify shield construction in the revised manuscript:

….the disturbance of shield construction, which is one of popular construction method in coastal cities of China….

in Fig. 3 increase font for the label of horizontal axis; given the importance of equation (21) increase font size?

Both the label of horizontal axis in Fig.3 and equation (21) have been increased in the revised manuscript.

the term nephogram was not familiar to me in this context (normally it refers to clouds) – perhaps clarify in parenthesis?

All the term nephogram have been replaced with clouds in the revised version.

Fig. 17 fonts for symbols are extremely small and hard for people with reduced vision to see

The symbols in both Fig.16 and Fig.17 have been revised in the revised version.

Reviewer 2 Report

Comments and Suggestions for Authors

Title: Constitutive Modeling of Physical Properties of Coastal Sand during
Tunnelling Construction Disturbance

Dear authors,

The topic of your paper is interesting. Overall, the quality of this paper is adequate.

It would be interesting to contrast the behavior of the model with physical parameters such as water content and/or mass density. However, your research is consistent.

Please, you should use line numbering throughout your paper.

Following are some recommendations for authors to consider:

A.- Abstract, title and references:
The aim is clear. The references are relevant and appropriate; however, it would be interesting to present some additional recent references.

B.- Introduction:
The research question is clearly outlined.

C.- M&M:

The variables are defined and neasured appropriately. Methods are valid and reliable.

D.- Results & Discussion:

Data in appropriate way. Tables are ok.

Figure 1 is not clearly visible.
Figure 3 would look better with a slightly larger size.
Figure 9, 10 & 14: title must start with capital letter.
Fig.10: "es nephogram..." typo?
Fig. 10-15: keys are not readable, too small

Eq. 17 & 21 would look better with a slightly larger size.

E.- Conclusions/ Implications:

Conclusions are supported by results or/and references.
However, it would be interesting some additional managerial implications in line with the findings of the study. Practical implications? Something to inspire future research or implications for practice.

Kind regards

Author Response

Reviewer2: The topic of your paper is interesting. Overall, the quality of this paper is adequate. It would be interesting to contrast the behavior of the model with physical parameters such as water content and/or mass density. However, your research is consistent.

Response: We agree with this reviewer. The authors will consider to incorporate the physical parameters such as water content, mass density etc into shield construction analysis and design in future investigations.

In Page 16; Lines 25-30 the following sentence have been added in the revised manuscript:

....It should be noted here that the proposed disturbance function is fundamental and does not cover physical parameters such as water content, mass density, etc....

Comment #2-1: Please, you should use line numbering throughout your paper.

Response #2-1: Thanks for reminding, the line number throughout of the present paper has been added in the revised manuscript.

Comment #2-2:  Abstract, title and references: The aim is clear. The references are relevant and appropriate; however, it would be interesting to present some additional recent references.

Response #2-2: Thanks for this advice. 5 recent references have been added in the revised manuscript. Moreover, one old reference published in 1986 has been removed in the revised version.

Comment #2-3: Introduction: The research question is clearly outlined. The variables are defined and measured appropriately. Methods are valid and reliable. Results & Discussion: Data in appropriate way. Tables are ok.

Response #2-3: Thanks for the reviewer’s diligent reviewing.

Comment #2-4: Figure 1 is not clearly visible.

Response #2-4: As suggested by the reviewer, Figure 1 has been redraw with higher resolution in the revised version.

Comment #2-5: Figure 3 would look better with a slightly larger size.

Response #2-5: As suggested by the reviewer, Figure 3 has been slightly enlarged in the revised version.

Comment #2-6: Figure 9, 10 & 14: title must start with capital letter.

Response #2-6: Thanks for reminding, the titles of Figure 9, 10 & 14 have been started with capital letter

Comment #2-7: Fig.10: "es nephogram..." typo?

Response #2-7: Thanks for reminding, Fig.10“stresses nephogram” has been revised to be “stress clouds”

Comment #2-8: Fig. 10-15: keys are not readable, too small

Response #2-8: Thanks for reminding, Figs.10-15 have been enlarged in the revised manuscript to assure that keys are readable.

Comment #2-9: Eq. 17 & 21 would look better with a slightly larger size.

Response #2-9: As suggested by the reviewer, Eq. 17 & 21 had been slightly enlarged in the revised version.

Comment #2-10: Conclusions/ Implications: Conclusions are supported by results or/and references. However, it would be interesting some additional managerial implications in line with the findings of the study. Practical implications? Something to inspire future research or implications for practice.

Response #2-10: This is a good suggestion. The authors have added the following sentences in the revised manuscript [Page 16, Lines 21-24, 25-30]:

….indicating that the developed model is capable of reproducing the mechanical behaviour of sandy soil during the whole process of shield construction and can be extensively applied for predicting the construction disturbance effect in practical tunneling engineering....

....It should be noted here that the proposed disturbance function is fundamental and does not cover physical parameters such as water content, mass density, etc. Moreover, the inherent relationship between the mechanical parameters such as stiffness, cohesion, internal friction angle, etc. of clay or sand-clay admixture and the aforementioned physical parameters under the disturbed state of tunneling construction should be further addressed. More advanced solutions including these parameters will be further studied in future research.….